# Effects of Temperature on Bending Properties of Three-Dimensional and Five-Directional Braided Composite

**DOI:** 10.3390/molecules24213977

**Published:** 2019-11-03

**Authors:** Peng Li, Na Jia, Xiaoyuan Pei, Zhenkai Wan, Jialu Li, Zhenrong Zheng, Hailiang Wu

**Affiliations:** 1School of Textiles Science and Engineering, TianGong University, Tianjin 300387, China; pengli@tjpu.edu.cn (P.L.); jiana@tjpu.edu.cn (N.J.); wanzhenkai@tjpu.edu.cn (Z.W.); lijialu@tjpu.edu.cn (J.L.); zhengzhenrong@tjpu.edu.cn (Z.Z.); 2Shanghai Composite Materials Technology Co., Ltd., Shanghai 201112, China; 3Composites Research Institute, TianGong University, Tianjin 300387, China; 4Dongfang Electric (Tianjin) wind power blade Engineering Co., Ltd., Tianjin 300480, China; wuhailiang@dongfang.com

**Keywords:** three-dimensional and five-directional braided composite, bending strength, high temperature, crackle

## Abstract

The bending properties of three-dimensional (3Dim) and five-directional (5Dir) braided/epoxy resin composites at room temperature, 90 °C, 110 °C, and 150 °C and heating for 0.25 h, 10 h, and 30 h, respectively, were studied. The effect of different temperatures and heating times on the bending property of these composites was discussed. The results showed that the bending strength of these composites at 90 °C, 110 °C, and 150 °C and heating time of 0.25 h is 33.86%, 46.27%, and 83.94% lower, respectively, than that at room temperature. In addition, 3Dim–5Dir braided composites exhibit different damage modes at different temperatures, revealing different failure mechanisms. Heating temperature has greater influence on the bending properties of these composites than heating time. The results provided a basis for the application of resin-based 3Dim–5Dir braided/epoxy resin composites at different temperatures.

## 1. Introduction

In the late 1960s, to meet the needs of the aerospace industry, a three-dimensional (3Dim) braided carbon/carbon composite was developed by the General Electric Company of the United States to replace superalloys for making rocket engine parts. This reduced the weight of rocket engines by 30%–50%. This successful attempt showed the development prospect of 3Dim braided composites, and the research of 3Dim braided composites had been raised. With continuous expansion of the applications of composite materials, the application temperature of resin matrix composites is required to be higher and higher [1,2]. The change of temperature will cause change to the fiber, resin, and interface in composites. This will affect the mechanical properties of composites [3]. Therefore, the research on the change of properties of resin matrix composites at different temperatures, especially in mechanical properties, electrical properties, and hygroscopicity, has attracted great attention [4,5,6]. 3Dim braided composites have 3Dim integral structure due to their reinforcement [7,8]. It breaks the traditional concept of composite laminate structure. Braided composites have tremendous advantages and potential for improving strength, impact resistance, fracture toughness, and damage tolerance between and within layers [9,10]. They are widely used in aviation, aerospace, and other high-tech fields.

In practical engineering applications, composites are often affected by a high-temperature environment [11]. This will cause stress and strain changes to the composites, thus affecting the normal working and mechanical properties of the composites [12]. Therefore, composites are required to have high environmental adaptability and maintain high structural stability despite drastic environmental changes. Compared with laminated composites, 3Dim–5Dir braided composites have better mechanical properties in the braiding direction due to added axial yarns. The axial yarns do not participate in the braiding, but they are surrounded by braided yarns. The inner ideal structure of 3Dim–5Dir braided fabric is shown in Figure 1a. The ratio of braided yarn and axial yarns is 1:1, and the arranged form of braided yarns and axial yarns on the chassis of braiding machine by rows and columns is shown in Figure 1b. In this paper, the structure of the specimen is a 3Dim–5Dir braided structure and the ratio of the braided yarns and the axial yarns is 1:1. However, the ratio of the braided yarns and the axial yarns could be different. For example, the ratio may be 2:1, 3:1, or 4:1, and the type and thickness of the axial yarns could be different from the braided yarns [13].

When the temperature of the environment changes, each part of the braided structure will expand or shrink. This will cause the stress and strain changes to the composites, thus affecting the normal work and mechanical properties of the composites. The effect of temperature makes damage evolution and failure behavior of composites complicated. Researchers have made some progress on the effect of temperature on the properties of braided composites. Using the experimental method, Li et al. explored the low-temperature (−196 °C) effect of quasi-static compressive properties and the high-temperature (55 °C, 75 °C, 100 °C, and 125 °C) effect of the quasi-static three-point bending properties of 3Dim braided glass fiber/epoxy resin composites [14,15]. The temperature effect of quasi-static compression and bending properties of 3Dim braided composites was significant. In a liquid nitrogen environment (−196 °C), the overall compressive properties of 3Dim braided composites were improved in axial, in-plane, and transverse directions compared with those at room temperature. Compression failure modes of 3Dim braided composites vary with loading direction and temperature. In a liquid nitrogen environment, the brittleness of compression failure was more significant. With the increase of temperature, the bending properties of 3Dim braided composites decreased. In a high-temperature environment, the bending strength and modulus of 3Dim braided composites was lower. The plasticity and softening characteristics of bending behavior were enhanced. The main failure modes were resin microcracks and interfacial debonding. In a normal-temperature environment, the failure morphology showed a brittle fracture. Pan et al. explored the thermal coupling properties of 3Dim braided basalt/epoxy composites under impact compression at different strain rates (1200–2400/s) in high/low-temperature fields (−140–210 °C) based on experimental methods and Meso-Structural models [16,17,18,19]. The results showed that when the temperature was lower than the glass transition temperature (Tg), failure occurred easily in the initial stage; the brittleness was obvious in the low-temperature field. Therefore, the different temperatures, strain rates, and loading directions would lead to different compression failure modes or shear failure modes. Pei et al. studied the compressive properties of 3Dim–5Dir braided composites at Tg of the polymer matrix. The matrix was TDE-86 epoxy resin with a Tg of 156 °C. At 150 °C, the compression strength retention rate of 3Dim–5Dir braided composites increased in the early stage and later decreased. The main reason was that the resin occurred post-curing, when heating time was extended, so that the temperature distribution in the material became uniform, stabilizing the structure of the material [20,21]. Wang et al. researched a numerical modeling on the compressive behaviors of 3-D braided carbon/epoxy composites with different braided angles under high temperatures at the microstructure level. It was found that the compressive failure changes from ductile to brittle when the braided angle changes from larger to smaller. The high temperature has almost no influence on the damage mode, but it leads to the failure modes becoming more ductile. The higher temperature and larger braided angle lead to the ductile behaviors of the 3D braided under longitudinal compression [22].

At present, there are few studies on the mechanical properties of three-dimensional braided composites at different temperatures. To further understand that the bending properties of 3Dim–5Dir braided/epoxy resin composites are affected by temperature, the bending properties of 3Dim–5Dir braided/epoxy resin composites at room temperature, 90 °C, 110 °C, and 150 °C, and heating for 0.25h, 10h, and 30h were studied. The bending properties of the composites at different temperature was discussed, and the damage mechanism was revealed. The results provided a basis for the application of 3Dim–5Dir braided composites at different temperatures.

## 2. Experimental Procedure

In the experiment, the specimens are resin-based 3Dim–5Dir braided carbon/epoxy resin composites. All specimens were prepared at the Institute of Composite Materials, TianGong University. Solidified composite adopts Resin Transfer Molding [23]. The carbon fibers used in resin-based 3Dim–5Dir braided composites are T700-12K carbon fibers. The resin is TDE-86 epoxy resin. The curing agent is 70# anhydride, and the catalyst is aniline. Tg is an inherent property of amorphous polymer materials. It is the macroscopic embodiment of the transformation of the movement form of polymer. It has a direct impact on the use performance and process performance of materials. The Tg of epoxy resin is 150 °C. Three heating temperatures were selected based on the Tg of matrix resin (90 °C and 110 °C < Tg, 150 °C = Tg). Aluminum alloy reinforcing sheets were pasted on both ends of the specimens. The specification and parameter of the specimens that were used for bending experiment are shown in Table 1.

The bending properties of the specimens were tested on the AG-250KNE universal material testing machine of SHIMADZU Company with thermostat. The bending properties of the specimens were tested with reference to GB/T 1449-2005. The calculation formula of bending strength is shown in Equation (1):(1)$$\sigma_{f}=\frac{3PI}{2bh^{2}}$$
$\sigma_{f}$ (MPa): bending strength (or the bending stress when the deflection is 1.5 times the specimen thickness); P (N): failure load (or the maximum load, or the load when the deflection is 1.5 times the specimen thickness); I (mm): span length; h (mm): Specimen thickness; b (mm): Specimen width.

The test temperatures were room temperature, 90 °C, 110 °C, and 150 °C. During the test, the temperature of the specimen is measured by the sensor. When the temperature reaches the design value and stabilizes for 0.25 h, the time is started, and the bending performance is tested after the design time is reached. Three specimens were tested at each temperature point.

## 3. Results

The bending properties of resin-based 3Dim–5Dir braided composites at different temperatures and continuous heating times were compared. All the specimens are No. 1 specimens under the test conditions, such as the shape of compression surface, side shape, electron microscopy, and deflection–load curve. The bending properties aided composites under different conditions were systematically analyzed. By contrast, the effects of different temperatures on the bending properties of braided composites and different heating time on the bending properties of braided composites at the same temperature were compared. The retention rates of bending properties at the same temperature and time were compared with those at room temperature.

### 3.1. Load–Deflection Curve of 3Dim–5Dir Braided Composites after Bending Test at 90 °C

Figure 2 shows the deflection–load curves of 3Dim–5Dir braided composites at room temperature and different durations at 90 °C. The curves show the cumulative process of structural damage. It is found that the bending strength of the material at 90 °C is lower than that at room temperature, and the bending strength of the material is also affected by the different time placed at 90 °C. The bending strength of specimens placed for 0.25 h at 90 °C is the smallest compared with that of specimens at the same temperature but with different durations. The bending strength of specimens placed for 10 h is the highest, and the bending strength of specimens placed for 30 h is the middle. The bending deflection–load curves of materials at room temperature, lasting for 0.25 h and 10 h at 90 °C show a linear relationship when they reach their maximum. There is a certain non-linear relationship between the deflection–load curve of the 3Dim–5Dir braided composites after they are placed at 90 °C for 0.25 h. However, the non-linear relationship between deflection and load is shown at the beginning of the bending test of 3D braided composites after being placed at 90 °C for 30 h. It has a plastic platform area. This indicates that the material exhibits a certain toughness due to the long duration for which it is placed at 90 °C.

### 3.2. Fracture Load of 3Dim–5Dir Braided Composites at 90 °C

The fracture load of 3Dim–5Dir braided composites at 90 °C is shown in Table 2. It can be seen from the table that the bending strength at 90 °C is lower than that at room temperature. After 10 h at 90 °C, the bending strength of the specimens is retained the best, followed by 30 h at 90 °C and 0.25 h at 90 °C. The specimens were placed at 90 °C for 0.25 h. Because of the slow heat transfer in high temperature, the temperature distribution of common polymer materials was uneven, and the mechanical properties of materials changed greatly. Therefore, the failure of resin resulted in the failure of the whole specimens. During the process of placing the specimen at 90 °C for 10 h, the heat distribution of the resin is uniform, and the structure of the material is stable. After that, the material under bending external force can be uniformly loaded, so the retention rate of bending strength is the highest. When placed for 30 h and subjected to high temperature for a long time, the internal molecular chains of the resin changed, so the bending strength of the resin was lower than that of the 10 h specimen, but the uniform structure of the resin was more stable than that of the 0.25 h specimen, so the bending strength of the resin was higher than that of the 0.25 h specimen. From the value of bending modulus, the modulus of material is close under four test conditions, which shows that the influence of temperature and duration on the bending modulus of 3Dim–5Dir braided composites is not regular, and the bending modulus of 3Dim–5Dir braided composites measured after 10 h of bending test is the opposite after being placed at 90 °C. it is higher than room temperature.

### 3.3. Load–Deflection Curve of 3Dim–5Dir Braided Composites after Bending Test at 110 °C

Figure 3 is the deflection–load curve of 3Dim–5Dir braided composites at room temperature and 110 °C. It can be seen intuitively that before reaching the maximum bending stress, the specimens subjected to bending stress at room temperature and 110 °C for 0.25 h show an obvious linear relationship, while the specimens lasted for 10 h and 30 h at the same temperature. The bending load–deflection curves of the specimens show an obvious non-linear relationship at the beginning, and the law of the deflection–load curves at the initial stage is similar. It is proved that the stress of specimens lasting for 10 h and 30 h at 110 °C is the same at the beginning: under the actions of high temperature and long time, the resin softens and the whole material shows a certain toughness. It can be seen from the figure that the bending strength of braided parts at 110 °C is lower than that at room temperature, and the bending strength of braided parts at the same temperature for 10 h is greater than that of the specimens lasting for 0.25 h, while the bending strength of the specimens lasting for 30 h is the smallest.

### 3.4. Fracture Load of 3Dim–5Dir Braided Composites at 110 °C

The fracture load of 3Dim–5Dir braided composites at 110 °C is shown in Table 3. From Table 3, it can be found that the retention rate of bending strength is the highest when the braided specimens are placed for 10 h at 110 °C. That is to say, the bending strength of braided specimens is higher than that of 0.25 h and 30 h after 10 h of work. However, there is no significant difference in the retention rate of bending strength after 0.25h, 10 h, and 30 h at 110 °C, which indicates that the effect of temperature on bending performance at 110 °C is much greater than that of time. From the bending modulus, it can be found that the bending modulus at 110 °C does not decrease much than that at room temperature, which indicates that the deformation degree of 3Dim–5Dir braided material under certain loads at high temperature will be greater than that of the material under the same load at room temperature, because the braided material will have the same load deformation at high temperature. With certain degree of toughness, the deformation will be larger.

### 3.5. Displacement–Load Curves of 3Dim–5Dir Braided Composites in Bending Tests at 150 °C

From Figure 4, it can be observed that the bending deflection–load curve at room temperature is linear, while the bending deflection–load curve at 150 °C is non-linear, and the bending strength at 150 °C is far less than that at room temperature. When the material is placed at 150 °C for 0.25 h, a plastic platform region appears in the deflection–load curve. With the increase of the deflection, the material shows a certain linear relationship. After the material reaches its maximum, the material slowly decreases stepwise.

### 3.6. Fracture Load of 3Dim–5Dir Braided Composites at 150 °C

The fracture load of 3Dim–5Dir braided composites at 110 °C is shown in Table 4.

The bending properties of 3Dim–5Dir braided composites were tested under seven test conditions, including normal temperature, 90 °C duration of 0.25 h, 10 h, and 30 h, and 110 °C duration of 0.25 h, 10 h, and 30 h. From the test results, it can be concluded that the bending strength of 3Dim–5Dir braided composites decreases with the increase of temperature. The bending properties of 3Dim braided materials with different durations at the same temperature are also different, but the bending strength of braided composites with durations of 10 h at 90 °C or 110 °C is the highest other than that at room temperature. However, the bending properties of 3Dim braided composites are less affected by different durations at 110 °C than at 90 °C. Because the bending strength of 3Dim braided materials with different durations at 110 °C is about 50%–60% compared with that at room temperature. The retention rate of bending strength is about 80%–90% under the test conditions of 10 h and 30 h at 90 °C, but the bending strength of braided materials with 0.25 h at 90 °C decreases by 66.14%. The bending strength of the material at 150 °C is only about 15% of that at room temperature. This also shows that the effect of temperature on the bending properties of braided materials is greater than that of duration on the bending strength of materials.

### 3.7. Comparison of Bending Properties of 3Dim–5Dir Braided Composites at Different Temperatures

The bending strength of 3Dim–5Dir braided composites heated at different temperatures for 0.25 h is shown in Table 5. From Table 5, it can be seen that the bending strength of 3Dim–5Dir braided composites decreases gradually with the increase of temperature, and the strength decreases obviously when the temperature rises to 150 °C. Compared to the bending strength at room temperature, the bending strength at 90 °C, 110 °C, and 150 °C degrees decreased by 33.86%, 46.27%, and 83.94% respectively. This shows that when the temperature is close to Martin’s heat-resistant temperature of epoxy resin, the bending strength of composites is significantly affected by temperature.

Figure 5 shows the deflection–load curves of the first braided composite specimen under bending test at room temperature and after heating at 90 °C, 110 °C, and 150 °C degrees for 0.25 h, respectively. The bending deflection–load curves of braided composites at room temperature are straight lines. When the material reaches the maximum load at 90 °C and 110 °C, the strength does not immediately decrease. The deflection–load curves show a certain non-linear relationship. This shows that the change of resin at high temperatures (90 °C and 110 °C) leads to the instability of the properties of braided composites, and shows a certain toughness after reaching the maximum value. The plastic plateau zone appears in the deflection–load curve of braided composites at 150 °C, which indicates that the resin softens at 150 °C, which makes the braided composites show some toughness.

## 4. Discussion

### 4.1. The Morphology of 3Dim–5Dir Braided Composites after Bending Test at 90 °C

Figure 6a–d is the appearance of the compressive surface (the surface in contact with the indenter is called the compressive surface) at room temperature and after 0.25 h, 10 h, and 30 h at 90 °C, respectively. It can be found that more specimens are broken and piled up at room temperature due to extrusion, which may be due to the softening of the resin at high temperature, and it can be observed that the phenomena of resin piled up by extrusion are more evenly distributed. The bending test specimens placed for 0.25 h at 90 °C are shown in Figure 6b. The phenomenon of resin accumulation is the least, but on the left side of the indentation there is some resin accumulation and peeling, and some fibers are warped. The specimens placed at 90 °C for 10 h and 30 h have cracks formed by resin accumulation and cracks on the edge of the specimens. However, the accumulation of specimens stored at 90 °C for 30 h is more obvious than that at the same temperature for 10 h.

Figure 7a–d are the side morphologies of 3Dim–5Dir braided composites after bending and fracture at room temperature, 90 for 0.25 h, 90 for 10 h, and 90 for 30 h, respectively. From Figure 7a, it can be found that there are obvious damage marks on the side of the bending fracture specimens at room temperature—cracks parallel to the forming direction, fine cracks at the interlacing of fibers, and resin fracture is also observed on the compression surface. It can be observed from Figure 7b that the resin fracture on the compression surface of specimens which lasted for 0.25 h at 90 °C was separated from the surface of specimens due to the bending stress on the upper surface. At the same time, the fibers slipped along the braiding angle and formed cracks. In Figure 7c, it can be found that braided pieces lasting for 10 h at 90 °C are subjected to bending action, and obvious cracks are formed at the interlaced fibers on the side of the braided pieces. In Figure 7d, a crack perpendicular to the bundle of fibers can be clearly seen in the figure. At the same time, a crack occurs in the middle of the bundle along the forming direction of the specimen.

In Figure 7a–d, the crack morphologies of 3Dim–5Dir braided composites are 600 times larger under scanning electron microscopy after bending fracture of the composites at room temperature and 90 °C for 0.25 h, 10 h, and 30 h, respectively. In Figure 8a, it can be found that many fibers break regularly and only a few fibers are pulled out. It shows that the resin–fiber bonded three-dimensional braided composites can withstand external loads well at room temperature. Most fibers break, resulting in large defects in the interior of the material, which ultimately leads to the failure of three-dimensional braided composites. Observation in Figure 8b shows that the debonding of resin and fibers is obvious when the 3Dim–5Dir braided composite is kept at 90 °C for 0.25 h. It can be seen not only that there are many block resins on the fibers, but also that the fibers in a bunch of fibers are separated from each other and thus layered. Figure 8c is the bending fracture of 3Dim–5Dir braided composites lasting for 10 h at 90 °C. The plastic deformation of the matrix can only be observed in the figure. The matrix yields and forms the shear band of the matrix. This is also due to the bending, tension and compression of the fibers during the bending failure process, and the extrusion deformation between the fibers. Plastic deformation caused by complex stresses from various aspects. From Figure 8d, we can find many fibers pulled out, because the specimens were placed at 90 °C for 30 h, the specimens showed obvious ductile fracture [24].

### 4.2. The Morphology of 3Dim–5Dir Braided Composites after Bending Test at 110 °C

Figure 9a–d show the compressive surface morphologies of 3Dim–5Dir braided composites subjected to bending fracture at room temperature, 110 °C for 0.25 h, 10 h, and 30 h respectively. Figure 9b is the upper form of bending fracture of braided composites lasting 0.25 h at 110 °C. Compared with the other three figures, in Figure 9b there are many obvious cracks, and the specimen is most seriously damaged. This is because the specimen suddenly enters a higher-temperature environment, and the unstable resin properties lead to the unstable performance of the whole material. The matrix on the upper surface is first cracked by extrusion, and then the bond between the fibers and the resin is destroyed. Finally, the fibers are extruded and the cracks expand rapidly. In Figure 9c, the graph shows the compressive surface of the material with 10 h bending fracture at 110 °C. Form Figure 9c, it can be found that there are only a few cracks in the graph, and the cracks are very small. When the material is at 110 °C for 10 h, the resin changes from an unstable state to a stable state, and the residual stress in the material is released to a certain extent, so the damage is not very serious. Figure 9d shows the upper surface morphology of the braided composites which were ruptured by bending stress after lasting for 30 h at 110 °C. The specimens also show that the surface peeling of the matrix and the fiber bundles under compressive stress is not as obvious as under normal temperature. This is due to the long duration of high temperature, the plastic deformation of the matrix is more obvious, the matrix softening, and the failure of the specimens is caused by the compressive stress of the fiber bundles under mutual extrusion and ultimate shear failure.

Figure 10a–d show the side morphologies of braided composites that are placed for 0.25 h at room temperature, 110 °C, 10 h, and 30 h after being bent, 30 times larger under a microscope. In Figure 10b, a crack perpendicular to the forming direction of the braided composites can be observed. Careful observation of the cracks does not show that the fibers break each other, and the resin does not break. This is because the resin is unstable at high temperature. When the specimen is subjected to bending force, the resin cannot transmit the force well. The fibers are handed over to each other, and the fibers press each other, resulting in the slipping and filing of the overlapping fibers. At the same time, cracks along the braiding angle can be observed on the cracked fiber bundles, and the phenomenon of separation between fibers can be observed under the electron microscope. From Figure 10c, it can be observed that the resin aggregates along the braiding angle of the fibers, and there are no obvious cracks on the compression surface. In Figure 10d, it shows that the 3Dim–5Dir braided composites, which are placed at 110 °C for 30 h, have many cracks along the yarn braiding angle when they are bent.

In Figure 11a–d, the crack morphologies of 3Dim–5Dir braided composites are 600 times larger under scanning electron microscopy after bending fracture of the composites at room temperature and 110 °C for 0.25 h, 10 h, and 30 h, respectively. From Figure 11b, it can be found that the fiber is separated from each other, and the debonding between the fiber and the matrix is accompanied by the matrix shedding at the same time. It can be seen that although the temperature action time of the 3Dim–5Dir braided composite is only 0.25 h at 110 °C, the effect of temperature on it is significant. From Figure 11c, it can be seen that the fibers are fractured under bending stress after being placed at 110 °C for 10 h, but the fracture surfaces are not regular, and the fibers between the fiber bundles are separated from each other. At the same time, some resin shedding phenomena are observed. In Figure 11d, only a small number of fibers are separated from each other. This may be because the fracture is formed at the intersection of two fibers, so that there are holes on the left side of the figure, which cannot be observed carefully.

### 4.3. The Morphology of 3Dim–5Dir Braided Composites after Bending Test at 150 °C

Figure 12a and b shows the compressive surface and lateral morphology of the composite after bending and fracture at 150 °C, respectively. From Figure 12a, it can be found that the resin exhibits network peeling and the fibers are separated from each other. From Figure 12b, it can be found that the fibers on the tension surface are broken by tension. From Figure 12a,b diagrams, it can be seen that the composite material is seriously damaged at 150 °C. The bending stress not only causes the fracture of the surface fibers, but also causes the serious damage of the internal fibers.

### 4.4. Damage Morphology and Failure Mechanism of 3Dim–5Dir Braided Composites at Different Temperatures

The reason for the change of bending strength can be analyzed by the morphology of specimens after bending and fracture. Figure 13 shows the morphology of the braided composite after heating for 0.25 h at different temperatures. From Figure 13a, it can be seen that the phenomenon of resin breaking and accumulation is obvious, but relatively uniform, indicating that at room temperature, the combination of fibers and resins is good, and both fibers and resins can well withstand bending stress. As can be seen from Figure 13b, the phenomenon of resin accumulation is the least; only the phenomenon of resin accumulation and peeling on the left side of the indentation, and some fibers are warped. As can be seen from Figure 13c, the phenomenon of resin accumulation is less and the fibers break more. This is because the specimen suddenly enters a higher-temperature environment, and the unstable resin properties lead to the unstable performance of the whole material. The matrix on the upper surface is first cracked by extrusion, and then the bond between the fibers and the resin is destroyed. Finally, the fibers are extruded and the cracks expand rapidly. Figure 13d shows that the resin is peeled off in a network shape, and the fibers are separated from each other. The most serious damage is found in the specimens.

Figure 14 is the morphology of braided composite specimens after bending fracture at different temperatures. At room temperature, there are small cracks along the braiding angle on the side of braided composites after bending fracture, while cracks along the braiding angle at 90 °C are more obvious as shown in Figure 14b. Figure 14c shows cracks along the thickness direction. The most serious failure of braided composites is at 150 °C. Resin peeling and fiber bundles break along the braiding angle.

Figure 15a–c show the electron microscopic pictures of side cracks of 3Dim–5Dir braided composite specimens after 0.25 h of heating at room temperature, 90 °C and 110 °C, respectively. From Figure 15a, it can be found that many fiber breaks are regular, and only a few fibers are pulled out. It shows that the resin–fiber bonded 3Dim braided composites can withstand external loads well at room temperature. Most fiber breaks result in large defects in the material. Failure of 3Dim braided composites. Figure 15b shows that the debonding phenomenon between resin and fiber is obvious when the 3Dim–5Dir braided material is retained at 90 °C for 0.25 h. It can be seen not only that there are many block resins on the fibers, but also that the fibers in a bunch of fibers are separated from each other, resulting in delamination. Figure 15c shows the separation between fibers, the debonding between fibers, and matrix accompanied by matrix shedding. It can be seen that temperature has a significant effect on the 3Dim–5Dir braided composites at 110 °C, although the time of temperature effect on the 3Dim–5Dir braided composites is only 0.25 h.

3Dim–5Dir braided composites are subjected to bending loads. The compressive surface (upper surface) of the specimen bears compressive stress and the tensile surface (lower surface) bears tensile stress. When the load on the specimen reaches a certain value, the compressive surface of the specimen first breaks down, showing collective plastic deformation of the resin, or even collective plastic deformation. The fibers on both sides of the indenter will bend and deform at the same time, and then break. With the increase of the load, the fibers will extrude each other to cause the shear failure of the fibers along the direction of braiding angle under the action of shear stress, which also causes the matrix failure. With the accumulation of damage, the interface between fiber bundle and matrix is destroyed gradually. Gradually, the bond between the fiber bundle and the matrix is destroyed [25,26]. At the same time, the fibers on the lower surface cannot support the fibers on the upper surface under tension, and the bundles will lose stability under compression and ultimately fail. Under the action of high temperature, the interface between fiber and resin was destroyed. The bond between the fiber and the resin became poor. Under the action of bending load, the resin peeled off from the fiber, and the fiber was pulled out. However, in all cases, there were no total fracture and separation of braided composite specimens. Because the reinforcement of 3Dim–5Dir braided composite was a whole structure. To a certain extent, the whole structure made up for the degradation of mechanical properties caused by resin damage at high temperature.

From the analysis of test results, the bending properties of braided composites decrease with the increase of temperature. The bending failure of specimens lasting for 0.25 h at 90 °C is the most serious and the bending strength is the smallest. Although the way of fiber bearing force is the same as that of specimens at room temperature, the residual stress in the material is generated after the material is cooled at high temperature when RTM (Resin Transfer Molding) is applied to the composite. When braided composites enter the high-temperature environment, the residual stress increases rapidly due to the movement of molecular chains, which leads to the inhomogeneity of the internal structure of the composites and the damage to the specimens. The bending properties of the composites decrease greatly within a short period of time after entering the high temperature. With the prolongation of holding time at high temperature, the residual stress is released due to the gradual relaxation of molecular chains. In a certain period of time, the material structure is more stable, and the bending properties are restored. Therefore, the bending performance of braided specimens placed at 90 °C for 10 h is higher than that of the same specimens placed at 90 °C for 0.25 h. After 30 h at 90 °C, the braided specimens have a longer time at high temperature, and the molecular chain of the resin is damaged, which makes the bonding force between the resin and the fiber worse [2]. Therefore, the bending performance of the three-dimensional braided composites is lower than that of the specimens at 90 °C for 10 h. Because its structure is more stable than that of specimens lasting for 0.25 h at the same temperature, the bending strength of specimens is higher than that of specimens lasting for 0.25 h at 90 °C.

The bending properties of 3Dim–5Dir braided composites with different durations at 110 °C are not as different as those of the samples at 90 °C, indicating that the durations at 90 °C have a greater impact on the braided specimens, while the bending properties of the braided specimens at 110 °C are not affected by the durations at 90 °C. The bending strength of specimens with 30 h at 110 °C is also smaller than that of specimens with 0.25 h, which is different from that of specimens with 90 °C. This may be due to the further influence of temperature on the properties of the resin, which leads to the decline of the overall properties of the material.

At 150 °C, the Tg of the resin is reached. Resin matrix is seriously damaged by high temperature. Because the 3Dim–5Dir braided reinforcement structure is an integral structure, it can support the specimen to bear considerable bending load to a certain extent. At this time, the combination of fiber and resin is poor. Therefore, when the bending load is applied, the resin is peeled off from the fiber. The fiber lost the binding effect of resin, there is separation between the bundles.

## 5. Conclusions

In this paper, the bending properties of 3Dim–5Dir braided composites were tested under seven test conditions, including normal temperature, 90 °C duration of 0.25 h, 10 h, 30 h, and 110 °C duration of 0.25 h, 10 h, and 30 h. From the test results, it can be concluded that the bending strength of 3Dim–5Dir braided composites decreases with the increase of temperature. The bending properties of 3Dim braided materials are different at different temperatures. However, the retention ratio of flexural strength of braided composites with a duration of 10 h at 90 °C or 110 °C is the highest at room temperature. The bending properties of 3Dim braided composites are less affected by different durations at 110 °C than at 90 °C. The retention ratio of bending strength of 3Dim braided materials under different durations at 110 °C is about 50–60% compared with that at room temperature. The retention rate of bending strength of specimens is about 80–90% under the conditions of 10 h and 30 h at 90 °C. The bending strength of the braided material, which lasted for 0.25 h at 90 °C, decreased by 66.14%. The retention rate of bending strength at 150 °C is only about 15% compared with that at room temperature. The effect of temperature on the bending properties of 3Dim–5Dir braided materials is greater than that of duration of heating. The results of this paper provide a reference for the application of resin-based 3Dim–5Dir braided composites at different temperatures.

## Figures and Tables

**Figure 1 molecules-24-03977-f001:**
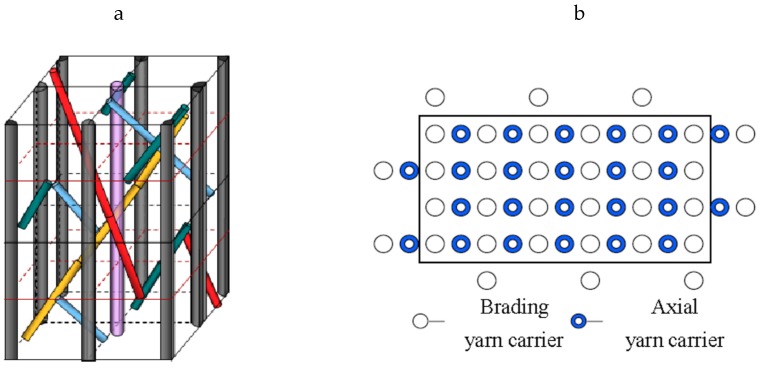
(**a**) The inner ideal structures of 3Dim–5Dir fabric. (**b**) Schematic illustration of carrier position on the machine bed.

**Figure 2 molecules-24-03977-f002:**
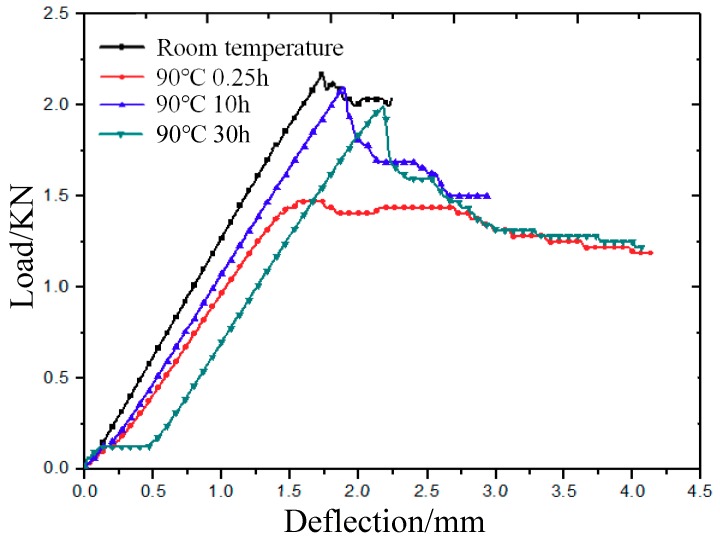
Bending deflection–load curve of 3Dim–5Dir braided composites at 90 °C with different durations.

**Figure 3 molecules-24-03977-f003:**
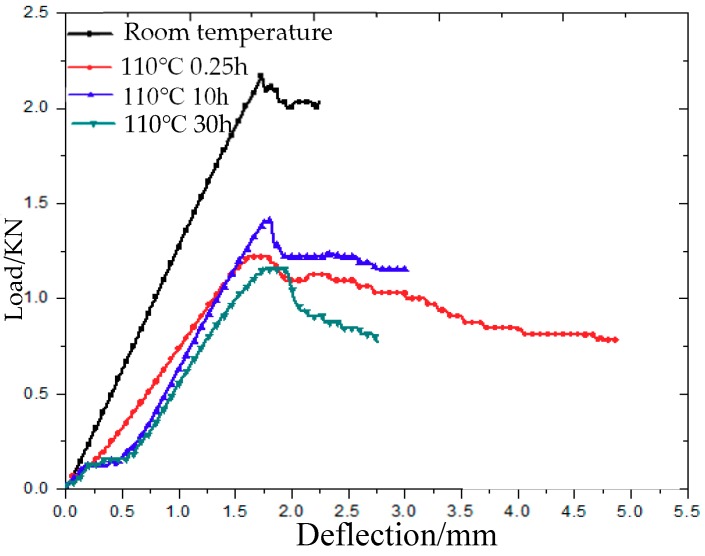
Deflection–load curve of 3Dim–5Dir braided composites after bending at 110 °C.

**Figure 4 molecules-24-03977-f004:**
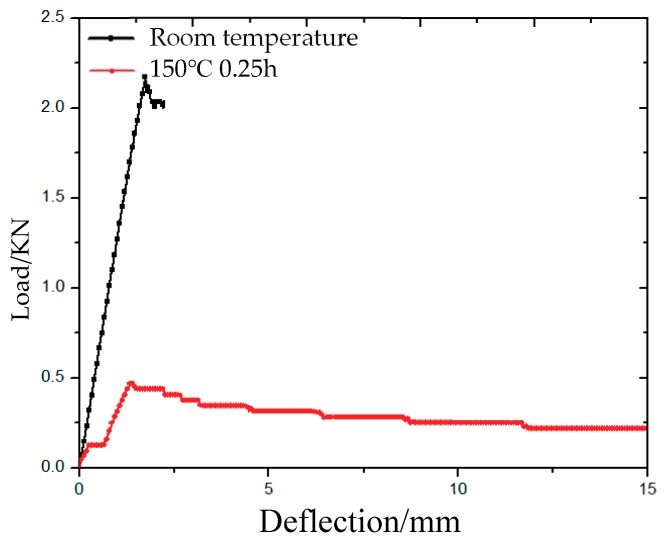
Deflection–load curves of 3Dim–5Dir braided composites at room temperature and 150 °C.

**Figure 5 molecules-24-03977-f005:**
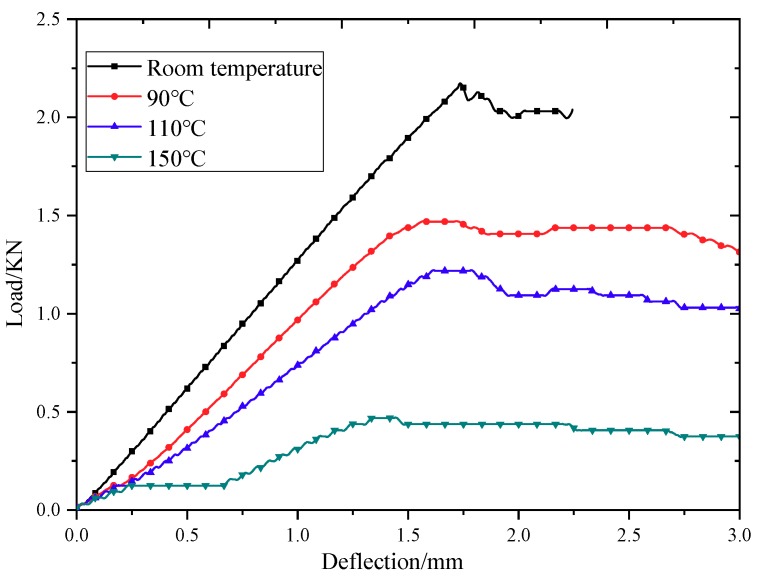
Bending deflection–load curves of 3Dim–5Dir braided composite specimens at different temperatures.

**Figure 6 molecules-24-03977-f006:**
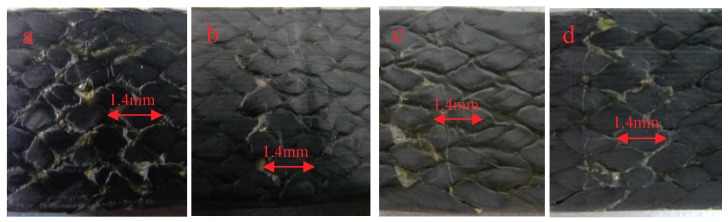
The compression surface morphology of 3Dim–5Dir braided composites with bending fracture at 90 °C and different durations (**a**) at room temperature (**b**) 90 °C 0.25 h; (**c**) 90 °C 10 h; (**d**) 90 °C 30 h.

**Figure 7 molecules-24-03977-f007:**
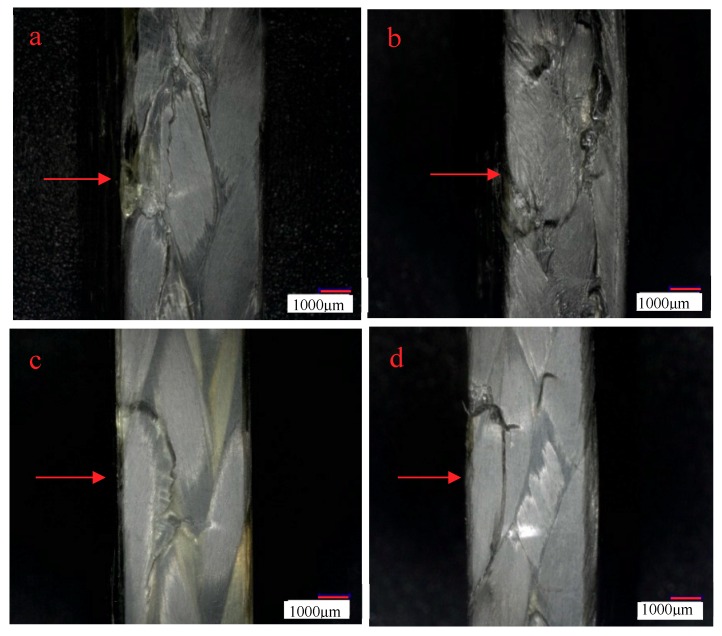
Lateral morphology of bending fracture of 3Dim–5Dir braided composites at 90 °C with different durations (**a**) at room temperature; (**b**) 90 °C 0.25 h; (**c**) 90 °C 10 h; (**d**) 90 °C 30 h.

**Figure 8 molecules-24-03977-f008:**
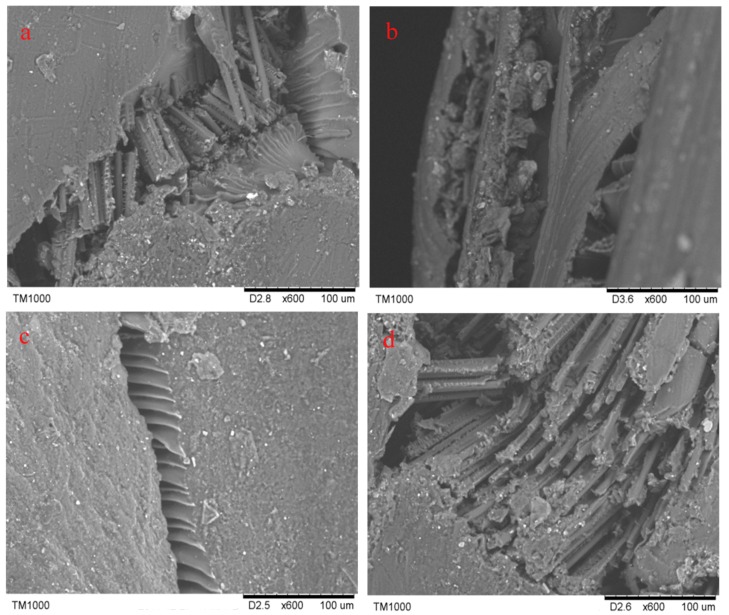
Electron micrograph of lateral crack after bending fracture in 3Dim–5Dir braided composites at 90 °C with different durations (**a**) at room temperature; (**b**) 90 °C 0.25 h; (**c**) 90 °C 10 h; (**d**) 90 °C 30 h.

**Figure 9 molecules-24-03977-f009:**
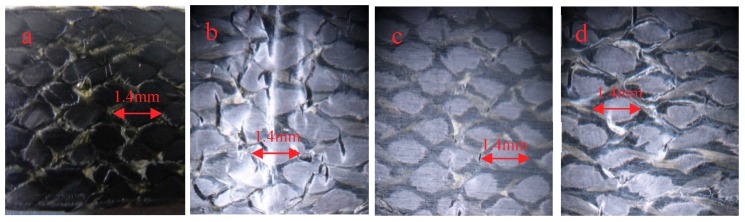
The compression surface morphology of 3Dim–5Dir braided composites with bending fracture at 110 °C and different durations (**a**) at room temperature; (**b**) 110 °C 0.25 h; (**c**) 110 °C 10 h; (**d**) 110 °C 30 h.

**Figure 10 molecules-24-03977-f010:**
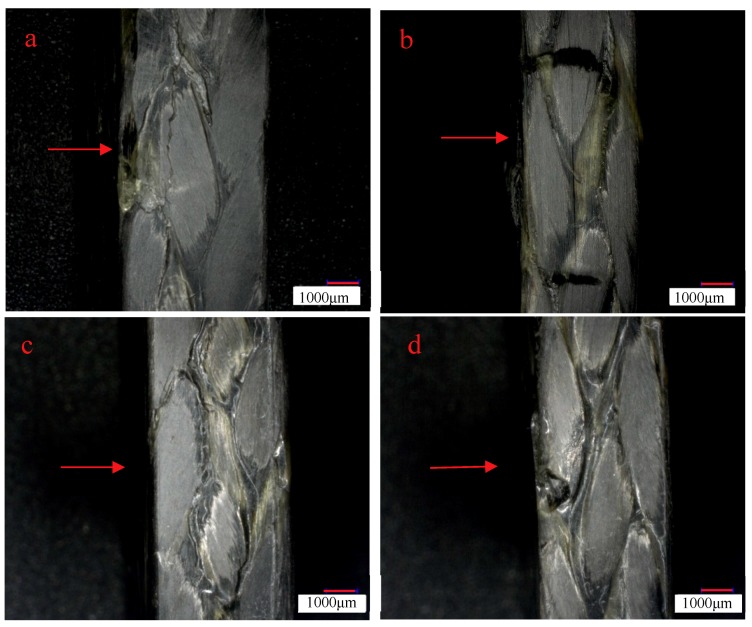
Lateral morphology of bending fracture of 3Dim–5Dir braided composites at 110 °C with different durations (**a**) at room temperature; (**b**) 110 °C 0.25 h; (**c**) 110 °C 10 h; (**d**) 110 °C 30 h.

**Figure 11 molecules-24-03977-f011:**
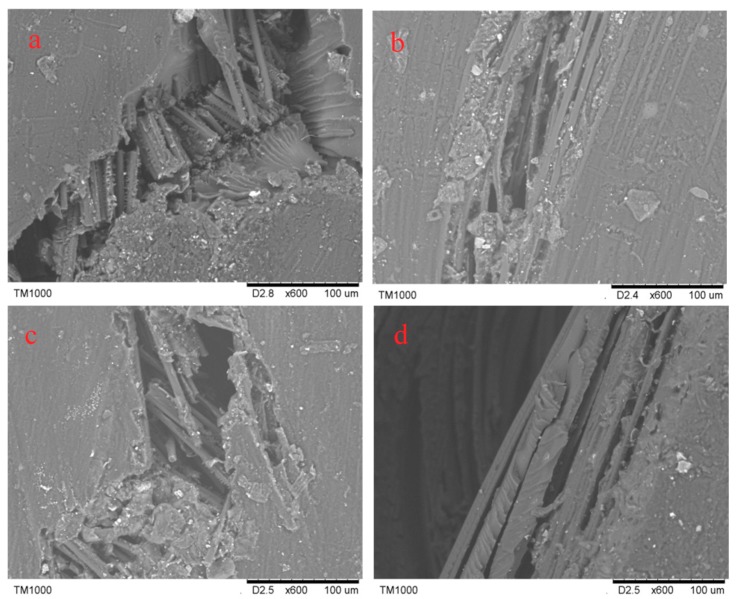
Electron micrograph of lateral crack after bending fracture in 3Dim–5Dir braided composites at 110 °C with different durations (**a**) at room temperature; (**b**) 110 °C 0.25 h; (**c**) 110 °C 10 h; (**d**) 110 °C 30 h.

**Figure 12 molecules-24-03977-f012:**
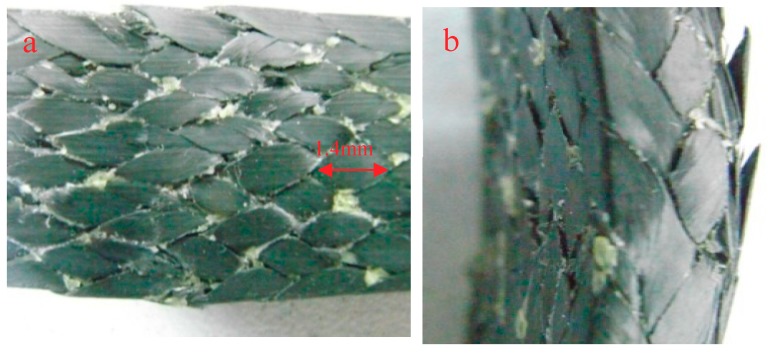
The bending fracture morphology of 3Dim–5Dir braided composites at 150 °C (**a**) at room temperature; (**b**) 150 °C 0.25 h.

**Figure 13 molecules-24-03977-f013:**
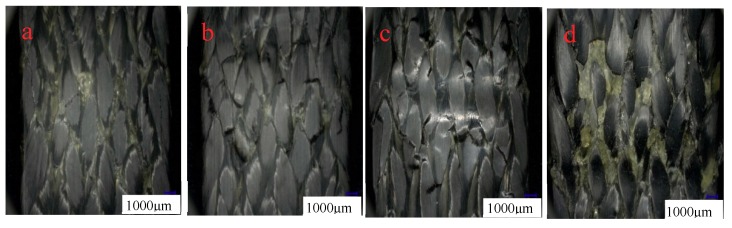
Pressure surface morphology of bending fracture of specimens at different temperatures (20 times magnification) (**a**) at room temperature; (**b**) 90 °C 0.25 h; (**c**) 110 °C 0.25 h; (**d**) 150 °C 0.25 h.

**Figure 14 molecules-24-03977-f014:**
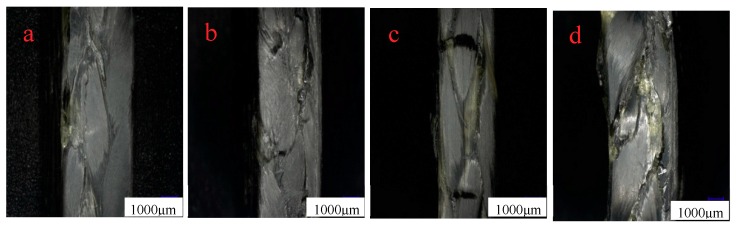
Lateral morphology of bending fracture of the specimens at different temperatures (30 times magnification) (**a**) at room temperature; (**b**) 90 °C 0.25 h; (**c**) 110 °C 0.25 h; (**d**) 150 °C 0.25 h.

**Figure 15 molecules-24-03977-f015:**
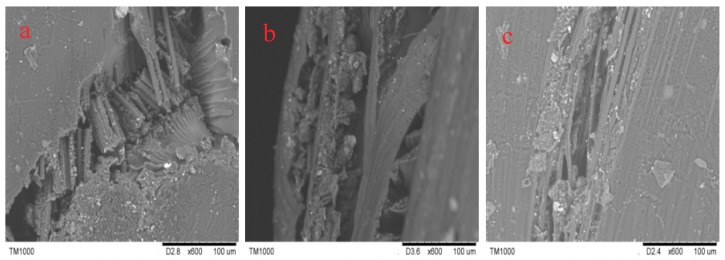
Electron microscopy of bending fracture of specimens at different temperatures (**a**) at room temperature; (**b**) 90 °C 0.25 h; (**c**) 110 °C 0.25 h.

**Table 1 molecules-24-03977-t001:** Specifications of composites specimen.

Specimen	Height × Wide × Thickness (mm^3^)	Surface Braiding Angle (°)	Fiber Volume Fraction (%)
3Dim-5Dir braided composite	80 × 15 × 4	29.6	54

**Table 2 molecules-24-03977-t002:** Bending properties parameters of 3Dim–5Dir braided composites at 90 °C.

Bending Condition	Room Temperature	90 °C/0.25 h	90 °C/10 h	90 °C/30 h
Maximum load/KN	2.25	1.49	2.08	1.93
Bending strength/MPa	776.23	513.42	718.07	666.42
Retention rate/%	-	66.14	92.51	85.85
Bending modulus/GPa	57.38	57.37	59.24	54.99

**Table 3 molecules-24-03977-t003:** Bending stress of braided structure specimens at room temperature and 110 °C.

Bending Condition	Room Temperature	110 °C/0.25 h	110 °C/10 h	110 °C/30 h
Maximum load/KN	2.25	1.21	1.48	1.10
Bending strength/MPa	776.23	417.05	509.83	380.57
Retention rate/%	-	53.73%	65.68%	49.03%
Bending modulus/GPa	57.38	43.01	50.64	36.52

**Table 4 molecules-24-03977-t004:** Bending stress of braided structure specimens at room temperature and 150 °C.

Bending Condition	Room Temperature	150 °C/0.25 h
Maximum load/KN	2.25	0.34
Bending strength/MPa	776.23	124.72
Retention rate/%	-	15.29%
Bending modulus/GPa	57.38	22.19

**Table 5 molecules-24-03977-t005:** Bending strength of 3Dim–5Dir braided composites at different temperatures.

Sample	Bending Strength/MPa			
	Room Temperature (23 °C)	90 °C	110 °C	150 °C
1	753.97	506.24	420.07	170.06
2	840.13	484.693	421.79	97.58
3	734.58	549.32	409.29	102.72
Average value	776.23	513.42	417.05	124.72

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
