# Peer review of "Effects of Temperature on Bending Properties of Three-Dimensional and Five-Directional Braided Composite"

_molecules, 2019, doi:10.3390/molecules24213977_

Round 1
Reviewer 1 Report
This paper presented an interesting study of the temperature and time effects on fiber-reinforced composites' mechanical properties. The authors first reviewed the state of the art, then detailed materials and experimental procedures were introduced. Adequate results were reported. The paper was well organized and easy to read. The reviewer suggests accepting this paper after a minor revision. Please see the detailed comments below.
Please include the manufacturer's recommended curing process for the epxoy resin and hardener used in this paper as baseline. This info should be compared to the curing time and temperature reported in this paper. The scale bars in Figs 3, 5, 7 were not readable. Please revise the images. Equations for bending strength calculations should be given in the revised paper.Author Response
please see the attachment

Reviewer 2 Report
This work presents and analyse experimental data of 3D reinforced composites tested at bending under high temperatures.
Although the manuscript contains interesting results, it needs extensive reformulation to be reconsidered for publication.
The text needs a careful editing process. Many phrases are very confusing. A paradigmatic example is in lines 77-80: "The results showed that when the glass transition temperature was lower than the glass transition temperature, the failure was easy to occur in the initial stage; the brittleness was obvious in the low temperature field, and the temperature rise was obvious in the failure stage."
The organization of the paper is inappropriate. It must separate the Results from Discussion to avoid excessive repetitions.
Give examples of application for this composite.
A detail explanation of the bending tests is necessary. Did it occur total separation of the broken specimens?
Provide a reference and a definition for "Martin's heat-resistant temperature".
The way used to describe the heating time prior to bending tests is inconvenient. Several figure captions fail to describe a), b),... and d) images. Some images omit the scale and others contain only a red line. In many photos is not a simple task to see the cracks described in the text.
The slope of bending deflection-load curves in figure 5 contradicts the results in Table 2.
The authors are somewhat ambiguous concerning hypothesis versus observed facts. An example is in lines 252-254: “…specimens have a longer time at high temperature, and the molecular chain of the resin is damaged, which makes…” This was observed or is an hypothetical explanation?
Line 377:” In this chapter, the bending properties of 3Dim-5Dir braided composites…” Why this chapter?
Author Response
Dear the editor and reviewers
Thank you for your valuable comments. We have revised our manuscript in line with your comments. The main revisions include:
Checking the manuscript carefully. Refining the language and correcting the grammatical errors and bad words and sentences as possible as we can. Some statements have been amended to improve the clarity.The comments are replied specifically as follows:
Response to Reviewer Comments
Point 1:
"The results showed that when the glass transition temperature was lower than the glass transition temperature, the failure was easy to occur in the initial stage; the brittleness was obvious in the low temperature field, and the temperature rise was obvious in the failure stage."
This sentence has been modified
The results showed that when the temperature was lower than the glass transition temperature (Tg), the failure was easy to occur in the initial stage; the brittleness was obvious in the low temperature field.
Point 2:
The organization of the paper is inappropriate. It must separate the Results from Discussion to avoid excessive repetitions.
The organization of the paper has been adjusted. Results and discussion has been separated. Duplicate parts have been removed.
Point 3:
Give examples of application for this composite.
In the late 1960s, in order to meet the needs of the aerospace field, the 3Dim braided carbon / carbon composite was prepared by General Electric Company of the United States to replace the superalloy to make rocket engine parts. This reduced the weight of the rocket engine by 30% - 50%. This successful attempt showed the development prospect of 3D braided composites, and the research climax of 3D braided composites had been raised.
Point 4:
A detail explanation of the bending tests is necessary. Did it occur total separation of the broken specimens?
Under the action of high temperature, the interface between fiber and resin was destroyed. The bond between the fiber and the resin became poor. Under the action of bending load, the resin peeled off from the fiber, and the fiber was pulled out. However, in all cases, there were no total fracture and separation of braided composite specimens. Because the reinforcement of 3Dim-5Dir braided composite was a whole structure. To a certain extent, the whole structure made up for the degradation of mechanical properties caused by resin damage at high temperature.
Point 5:
Provide a reference and a definition for "Martin's heat-resistant temperature".
In this paper, the glass transition temperature of resin is wrongly writed as Martin's heat-resistant temperature, which is now corrected.
Tg is an inherent property of amorphous polymer materials. It is the macroscopic embodiment of the transformation of the movement form of polymer. It has a direct impact on the use performance and process performance of materials. The Tg of epoxy resin is 150 ℃. Three heating temperatures were selected based on the Tg of matrix resin (90℃/110℃<Tg, 150℃=Tg).
Point 6:
The way used to describe the heating time prior to bending tests is inconvenient. Several figure captions fail to describe a), b),... and d) images. Some images omit the scale and others contain only a red line. In many photos is not a simple task to see the cracks described in the text. The slope of bending deflection-load curves in figure 5 contradicts the results in Table 2.
Thanks for the reviewer valuable comments.
As for the image scale, the problem of unclear crack description in the photo, the problem of the slope of bending deflection load curve in Figure 5 and the result in Table 2 had been all corrected.
Point 7:
The authors are somewhat ambiguous concerning hypothesis versus observed facts. An example is in lines 252-254: “…specimens have a longer time at high temperature, and the molecular chain of the resin is damaged, which makes…” This was observed or is an hypothetical explanation?
Thanks for the reviewer valuable comments.
“…specimens have a longer time at high temperature, and the molecular chain of the resin is damaged, which makes…” This is a hypothetical explanation, it was obtained by reference.
Point 8:
Line 377:” In this chapter, the bending properties of 3Dim-5Dir braided composites…” Why this chapter?
Thanks for the reviewer valuable comments. It has been corrected.

Round 2
Reviewer 2 Report
Although the overall quality of the revised manuscript improved, few issues remain:
a) Figure captions must describe a), b),... and d) images;
b) The separation of Results from Discussion was not done as advised.
